# Analysis of the Impact of Interpolation Methods of Missing RR-intervals Caused by Motion Artifacts on HRV Features Estimations

**DOI:** 10.3390/s19143163

**Published:** 2019-07-18

**Authors:** Davide Morelli, Alessio Rossi, Massimo Cairo, David A. Clifton

**Affiliations:** 1Institute of Biomedical Engineering, Department of Engineering Science, University of Oxford, Oxford OX2 6DP, UK; 2Biobeats Group LTD, 3 Fitzhardinge Street, London W1H 6EF, UK; 3Computer Science Department, University of Pisa, Largo Bruno Pontecorvo 3, 56127 Pisa, Italy

**Keywords:** heart rate, IoT wearable monitor, health

## Abstract

Wearable physiological monitors have become increasingly popular, often worn during people’s daily life, collecting data 24 hours a day, 7 days a week. In the last decade, these devices have attracted the attention of the scientific community as they allow us to automatically extract information about user physiology (e.g., heart rate, sleep quality and physical activity) enabling inference on their health. However, the biggest issue about the data recorded by wearable devices is the missing values due to motion and mechanical artifacts induced by external stimuli during data acquisition. This missing data could negatively affect the assessment of heart rate (HR) response and estimation of heart rate variability (HRV), that could in turn provide misleading insights concerning the health status of the individual. In this study, we focus on healthy subjects with normal heart activity and investigate the effects of missing variation of the timing between beats (RR-intervals) caused by motion artifacts on HRV features estimation by randomly introducing missing values within a five min time windows of RR-intervals obtained from the nsr2db PhysioNet dataset by using Gilbert burst method. We then evaluate several strategies for estimating HRV in the presence of missing values by interpolating periods of missing values, covering the range of techniques often deployed in the literature, via linear, quadratic, cubic, and cubic spline functions. We thereby compare the HRV features obtained by handling missing data in RR-interval time series against HRV features obtained from the same data without missing values. Finally, we assess the difference between the use of interpolation methods on time (i.e., the timestamp when the heartbeats happen) and on duration (i.e., the duration of the heartbeats), in order to identify the best methodology to handle the missing RR-intervals. The main novel finding of this study is that the interpolation of missing data on time produces more reliable HRV estimations when compared to interpolation on duration. Hence, we can conclude that interpolation on duration modifies the power spectrum of the RR signal, negatively affecting the estimation of the HRV features as the amount of missing values increases. We can conclude that interpolation in time is the optimal method among those considered for handling data with large amounts of missing values, such as data from wearable sensors.

## 1. Introduction

In the last two decades, the interest in the variation of the timing between beats (RR-intervals) of the cardiac cycle, called heart rate variability (HRV), has widely increased in the psycho-physiological research field. Assessment of RR-intervals variability is possible through time and frequency domain analyses that provide parameters able to quantify the amount of fluctuations occurring between consecutive beats, giving therefore an indirect index of autonomic regulation. Actually, the parameters extracted from HRV analysis are useful to provide insight about sympathetic-parasympathetic balance of cardiac vagal tone that was found to be an indicator of cognitive, emotional, social and health status [1].

Thanks to the technological advancements of recent decades, it is now possible to continuously record heart activity during peoples’ life via wrist-worn wearable devices equipped with heart rate sensors. This innovation might have a great impact on the medical field because of the low cost of the devices and the possibility to obtain continuous passive measurements performed in an ecological setting, gaining an overview of the users’ health status by assessing HRV features during their daily life [2]. These wrist-worn wearable devices, however, produce several inconsistent RR-intervals produced not only by ectopic beats (e.g., atrial fibrillation and premature heart beat), but mainly by motion and mechanical artifacts induced by external stimuli. The number of abnormal RR-intervals increases from 1%—when heart beats are recorded with gold standard technology (i.e., electrocardiography)—[3] to more than 10%—when they are recorded with wrist-worn wearable devices. However, standard methods for calculating HRV features from the time-series of RR-intervals require accurate beat detection. Hence, handling the missing values became a fundamental aspect to correctly evaluate users’ physiological response. As a matter of fact, these missing values affect the HRV analysis producing misleading results [4]. In previous studies, the inconsistent RR-interval data were handled by reconstructing the missing values using nearest-neighbour, linear, cubic spline and piecewise cubic Hermite interpolation methods [4,5]. However, these methods can also introduce changes in the reconstructed timeseries that could corrupt the signal spectrum [6], thus reducing the ability to estimate both time or frequency domains HRV features.

In this paper, we focus on healthy subjects with normal heart activity, and investigate the effects of interpolation on time (i.e., the timestamps when the heartbeats happen) and duration (i.e., the duration of the heartbeats) with an increasing amount of missing values (from 0% to 70%) in order to assess which interpolation strategy yields better results when estimating HRV features. In particular, in this paper we show that quadratic interpolation on time is the best approach to reconstruct the missing RR-intervals. Anyway, the main finding of this study is that the interpolation on time produce better HRV feature estimation that the interpolation on Duration suggested by all the previous studies.

### 1.1. Paper Contribution

To the best of our knowledge, this work is one of the first studies investigating the effect of high percentage of missing values (i.e., 30%, 50% and 70%) on HRV analysis. In previous studies, the inconsistency of RR-intervals was due to a small number of ectopic beats, while wrist-worn wearable devices introduce motion and mechanical artifacts that produce a huge quantity of abnormal heart beats.

Moreover, to the best of our knowledge, this is the first study to analyse the effect on HRV features of interpolation on time versus interpolation on duration. We show the difference among interpolation methods (i.e., no-interpolation, nearest neighbor, linear, quadratic and cubic spline) on both time and duration timeseries in order to detect which interpolation method yields lower error in HRV features estimations. This analysis permits to provide insight about how the interpolation methods work in quantifying the noise introduced into the timeseries.

We conclude by showing that interpolation on time is the best choice for preprocessing RR timeseries with missing values, contradicting the approach traditionally followed, based on durations timeseries.

### 1.2. Related Work

During the day, approximately 1% of beats are to be expected to be ectopic [3] when they are recorded by using gold standard instrument (i.e., Electrocardiography). An ectopic beat is a disturbance of the cardiac rhythm that induces premature ventricular or atrial contraction. The physiological artifact producing inconsistent beat seriously affects the HRV spectrum, and could result in erroneous results during HRV analysis by introducing non-existing frequencies into the spectrum [7]. In addition to physiological artifact, motion and mechanical artifacts induced by external stimuli introduce a large amount of inconsistent beats when the data are recorded by using wrist-worn wearable device [4,5,6]. This work is one of the firsts studies that investigate the effect of huge quantities of inconsistent beats that are not only derived from ectopic beats. Since missing data are common in the RR-interval timeseries derived from wrist-worn wearable device, they could complicate the analysis of HRV features making it sometimes impossible. To make reliable HRV analysis, previous studies suggested several preprocessing methods for RR-intervals timeseries (e.g., deletion, interpolations and filtering). However, these preprocessing methods have their own distinct effect on HRV analysis yielding different results [7].

The simplest way of handling the inconsistent RR-intervals provided in literature is to delete them [8]. In this approach, the abnormal RR-intervals are removed and the normal RR-intervals list are merged together. A huge issue of the deletion approach is that it reduces the overall length of the HRV signal. This may significantly influence HRV spectrum [8]. Other interpolation methods maintain the original number of samples, but, by manipulating the duration of RR-intervals, they also change the overall duration by some amount. There are several interpolation approaches useful for handling inconsistent RR-intervals, i.e., zero degree, linear and cubic spline [9]. Zero degree replaces the inconsistent RR-intervals with the mean of the closest normal values. Differently, linear interpolation fits a straight line over the inconsistent RR-intervals to obtain normal values. Finally, the most popular interpolation approach is the spline of order three (i.e., cubic spline). It fits a third degree polynomial smooth curve through a number of data points to obtain new values. This latter approach is recommended when there is only small number of inconsistent RR-intervals [9].

Finally, it was found that the interpolation introduces low frequency components (LF) and reduces high-frequency components (HF) power [6]. This aspect affects frequency domain HRV features [5], while little effect was found in time domain HRV features [4].

We were not able to find any previous work studying the effect of interpolation missing values on the duration versus time, and the propagation of error to HRV features.

## 2. Materials and Methods

### 2.1. Dataset

In this paper, we used *nsr2db* (Normal Sinus Rhythm RR Interval Database) PhysioNet dataset [10]. This dataset contains beat annotations of 54 normal sinus rhythm subjects (30 men: 28–76 years; 24 women: 58–73 years) extracted from 23 h long electrocardiogram (ECG) recordings, digitized at 128 samples per second, and beat annotations obtained by automated analysis with manual review and correction.

In order to compute HRV features, the 23 h time series of ECG recording of each user were split into 5 min windows. Moreover, to investigate the effect of missing values on HRV analysis, artificial missing RR-intervals (i.e., 30%, 50% and 70% of missing values) were inserted into the 5 min windows.

The missing values were created in accordance with a burst Gilbert model that simulates burst-error with a two-state Markov chain (i.e., good as 0 and bed as 1) [11]. We define *P* as the probability of transition form state 0 to the state 1 and *p* the probability of transition from state 1 to 0. Moreover, *Q* and *q* give the probabilities of remaining in the same states 0 or 1 (see Figure 1). Using these parameters, it is possible to represent average bit-error rate $P_{e}$ as showed in Equation (Equation 1) and the average burst length ($L_{length}$) is set at 10.
(1)$$P_{e}=\frac{P}{p+P}.$$

Given these equation, we define *P*, *p*, *Q* and *q* as showed in Equations (Equation 2)–(Equation 5), respectively.
(2)$$p=1/L_{burst}$$
(3)$$P=\frac{P_{change}}{1-P_{change}}*p$$
(4)q=1−p
(5)Q=1−P,
where the $P_{change}$ is set in accordance with the missing values percentage that we want to add in the time series (e.g., if we want 20% of missing values we set $P_{change}$ as 0.3). The missing values were introduced in the time series when the state of the two-state Markov chain is equal to 1. Examples of 30%, 50% and 70% of missing values created by Gilbert model are provided in Figure 2.

### 2.2. Missing Values Interpolation

The missing values were then handled with six different interpolation methods:No interpolation: this approach does not create interpolated values of missing RR-intervals. Differently to the Deletion method that remove missing values merging non-consecutive beats that induce in missing interpretation of HRV features, the no-interpolation method maintains the missing values into the RR-intervals time series.Nearest neighbor: the nearest neighbor or proximate interpolation is the easiest interpolation method [12]. This interpolation assigns the value of the closest known (existing) neighbor to the missing- value as shows in Equation (Equation 6).
(6)$$X_{i}=\left\{\begin{array}{cc} x_{B} & \mathrm{if}\;i<\frac{a+b}{2}\\ x_{A} & \mathrm{if}\;i\geq \frac{a+b}{2} \end{array}\right.$$
where *a* and *b* are the indexes of $x_{A}$ and $x_{B}$. Interpolated data by this method are discontinuous and it often yields the worst results [13]Linear: this method fits a straight line passing through points $x_{A}$ and $x_{B}$ [14]. Interpolated data by the linear model are bound between $x_{A}$ and $x_{B}$ as showed in Equation (Equation 7).
(7)$$X_{i}=\frac{x_{A}-x_{B}}{a-b}\left(i-b\right)+x_{B}.$$
Gaunck et al. [14] demonstrated that this method is efficient, and most of the time it is better than non-linear interpolations for predicting missing values in environmental phenomena with constant rates. In addition, they also found that in average this interpolation model underestimated the real values but it strongly depends on the distribution of the data.Quadratic: differently from the linear interpolation model, the quadratic function needs three points of interest to interpolate missing values in a time series as showed in Equation (Equation 8).
(8)$$X_{i}=x_{B}\frac{\left(i-b\right)\left(x_{C}-x_{A}\right)}{2(b-a)}+\frac{{\left(i-b\right)}^{2}\left(x_{A}-2x_{B}+x_{C}\right)}{2{\left(b-a\right)}^{2}}.$$
Compared to the linear model, quadratic interpolation is found to be in general more accurate [13].Spline cubic: fitting datapoints using polynomials of degree higher than one leads to problems of oscillation outside the fitted points, known as Runge’s phenomenon [15]. This problem can be avoided by using a spline, a function defined piecewise by polynomials, using datapoints as control points instead of forcing the fitted function to pass through the data points. Cubic spline is a spline composed of piecewise third-order polynomials. By using third degree polynomials is possible to ensure that the resulting curve is smooth [15], avoiding the problem of the straight polynomial interpolation that tends to induce distortions on the edges of the polynomials, given by the fact that, in general, the first and second derivative of the function defined by piecewise polynomials will not be continuous at the edges of polynomials. With cubic spline, it is possible to force the first and second derivatives of consecutive polynomials to be equal, ensuring smoothness of the resulting curve.

We applied each of the interpolation methods listed above to heartbeats expressed as a sequence of durations and as a sequence timestamps, then analyzed the error in HRV features estimations, in order to identify the best approach.

The on-duration approach is the one mostly used in literature to handle missing values. The data used as input to the interpolation methods was the sequence of durations of the heartbeats (the RR-intervals), obtained by subtracting the timestamp of each heartbeat from the timestamp of the subsequent heartbeat in the sequence of heartbeats.

Differently, we propose the the on-time approach whereby interpolation methods are applied to the sequence of timestamps of the heartbeats, postponing the differentiation preprocessing step that transforms timestamps into durations to after the interpolation step.

As shown in Figure 3 the difference between the on-time and on-duration approaches is the order of the processing steps: in the on-duration approach the timestamps are converted to durations as the first processing step; in the on-time approach this step is performed after interpolation is performed.

To better illustrate the differences between interpolation on time and duration, we simulated 100 heartbeats and then we randomly generated 10% of missing RR-intervals in this artificial timeseries by using Gilbert burst approach. The length of the RR-intervals timeseries changes when we interpolate the missing values on duration, while it remains the same when we interpolate on time (Table 1). This result suggests that the interpolation on duration moves beats away from their original position in time, introducing changes to the spectrum, while interpolation on time preserves the position on time of retained heartbeats. In particular, Table 1 shows that the low RR-intervals error (i.e., average difference between heartbeats duration) is obtained with linear interpolation on time. The nearest interpolation on time was not performed because interpolating with this approach is useless, as the interpolated values introduced into the timeseries would have the same time as the closest beat, creating physiologically impossible data.

Figure 4 provide more detailed analyses of the difference between linear interpolation on both duration and time. The cumulative error when the missing values are interpolated on duration increases as the time series goes by because it creates RR-intervals in accordance with the closest interval values (i.e., the higher is the number of missing values, the higher is the cumulative error) depending on the interpolation type used (e.g., linear, quadratic and cubic spline). Differently, time interpolation did not introduce change in time series length due to the fact that this approach estimates intermediate values between the time when two observed beats happen in accordance with interpolation type.

### 2.3. Feature Engineering

To obtain HRV features, we analyzed real (i.e., without missing values values), non-interpolated, and interpolated (i.e., with different percentage of artificial missing values values) 5 min ECG time series. We analyze time domain HRV features, frequency, and non-linear domains. Time domain analysis usually contains various statistical variables of the duration time series. The frequency domain analysis investigates the power spectrum of RR-intervals time series in order to assess the cardiac autonomic balance (i.e., sympathetic and parasympathetic nervous systems activity). Additionally, non-linear HRV features try to capture the non-periodic behaviour of the HRV and the complexity that exists inside the RR-interval dynamics. The variables that we incude in our analysis, in both time and frequency domain, are defined as:Time domain:
-HR mean: mean values of heart rate (HR) computed as showed in Equation (Equation 9).
(9)$$HR_{mean}=\frac{1}{N-1}\sum_{i=1}^{N-1}60/\left(R_{i+1}-R_{i}\right),$$
where N is the number of beats and R is the time when the beats happened.-RMSSD: root mean square of the successive RR-intervals differences (Equation (Equation 10)) represents the strength of the autonomic nervous system (specifically the parasympathetic branch) at a given time.
(10)$$RMSSD=\sqrt{\frac{1}{N-1}\sum_{i=1}^{N-1}{\left[\left(R_{i+1}-R_{i}\right)-\left(R_{i}-R_{i-1}\right)\right]}^{2}},$$
where N is the number of beats and R is the time when the beats happened.-SDNN: standard deviation of RR-intervals (Equation (Equation 11)). It reflects the cyclic components responsible for variability in the RR-intervals time series. The SDNN is the “gold standard” for medical stratification of both morbidity and mortality [16].
(11)$$SDNN=\sqrt{\frac{1}{N-1}\sum_{i=1}^{N}{\left(RR_{i}-\bar{RR}\right)}^{2}},$$
where N is the number of beats and RR is the intervals between two consecutive R and $\bar{RR}$ is the mean of RR-intervals in the time series.-PNN50: the ratio between NN50 (i.e., number of pairs of successive RR intervals that differ by more than 50 ms) and the total number of RR-intervals (Equation (Equation 12)).
(12)$$PNN50=\frac{NN50_{count}}{N_{RR-intervals}}$$Frequency domain:
-Power spectral density (PSD): describes the distribution of power into frequency components composing that signal. The Lomb–Scargle periodogram for PSD estimation was found to be the most appropriate method to analyze RR-interval data [5,6]. VLF (power in very-low-frequency ranges, i.e., ≤0.04 Hz), LF (power in low-frequency ranges, i.e., 0.04–0.15 Hz), HF (Power in high-frequency ranges, i.e., 0.15, 0.4 Hz), LF/HF ratio (ratio between LF and HF expressed as ms${}^{2}$), and total power (Power in all the frequency ranges, i.e., ≤0.4) were obtained by the sum of the power in the relevant frequency range in the spectrum.Non-linear HRV features:
-Poincaré plot: it is a type of recurrence plot used to quantify self-similarity in processes. A Poincaré plot is a graph of RR interval ($RR_{n}$) against the previous one ($RR_{n}-1$). From this scatter plot, it is possible to quantitatively analyze the variance of two consecutive RR-intervals by fitting an ellipse to the plotted shape. SD1 is the standard deviation of Poincaré plot perpendicular to the line-of-identity, while SD2 is the standard deviation of the Poincaré plot along the line-of-identity.

### 2.4. Success Metrics

We assessed the difference of HRV variables computed on real time series and the ones with missing values by the root mean squared error (RMSE). Additionally, the relative errors (REs, see Equation (Equation 13)) were used to assess the effects of the missing data on the HRV features compared with the parameters calculated from the RR-intervals timeseries without missing data.
(13)$$RE=\frac{|x_{real}-x_{k}|}{x_{real}}*100,$$
where $x_{real}$ refers to the HRV features computed from RR-intervals timeseries without missing values, while $x_{k}$ refers to the values obtained from interpolated timeseries.

## 3. Results and Discussions

### 3.1. Results Summary

We analyzed 15,359 RR-intervals timeseries of 5 min in this study. Table 2 shows the descriptive statistic of the HRV features extracted from all users in the dataset. In particular, the users shows an average heart rate of about 75 ± 14 beats per minute.

Table 3 shows that the on-time approach (i.e., interpolation on the timestamp of heartbeats) produces more reliable HRV feature estimations compared to the on-duration approach (i.e., interpolation on interval duration between two consecutive heartbeats). In this table we provide the results of the best interpolation approach for each HRV feature and for all the percentages of missing values. The RE and RMSE values provided in this table refer to the error induced by missing values when we compare HRV features obtained from the real RR-intervals timeseries versus the ones obtained from interpolated timeseries. The best interpolation methods provided in Table 3 refer to the ones with lower RE. For all of the HRV features, the highest was the percentage of missing RR-intervals, and also the parameters estimation errors. This was due to the fact that the power spectrum of the RR-intervals signal changes with the number of missing values. The choice of the interpolation method also added different types of noise to the signal. As shown in Table 1, the interpolation on time, or not interpolation at all, produces more reliable HRV features compared to interpolating on duration.

The lowest errors on HRV features estimation with missing RR-intervals are obtained using the no-interpolation or the interpolation on time approaches, while the interpolation on duration approach consistently yields the worst results (Table 3). Even if low timeseries difference were detected in simulated linear interpolation on time (Figure 4 and Table 1), Table 3 suggests that the best interpolation method depends of the HRV features that we want to assess. Moreover, this table also shows that, as suspected, the higher is the percentage of missing values, the higher is also the HRV feature estimation error (i.e., RE and RMSE).

### 3.2. HRV Features

#### 3.2.1. Time Domain

RMSSD and PNN50 do not require any interpolation to obtain reliable estimations for all the percentages of missing values, while SDNN need quadratic interpolation on time (see Table 3). A possible explanation of this result is that RMSSD and PNN50 capture fast changes in heart activity, i.e., high spectrum frequencies, and SDNN captures slow changes, i.e., very low spectrum frequencies. Moreover, interpolation methods, especially interpolation on duration, act as low pass filters, affecting the signal measured by the HRV features (Figure 5). No interpolation changed the spectrum, but did not introduce fictuous durations, thus minimizing the impact on successive differences of durations, that were the first computation step of both RMSSD and PNN50.

#### 3.2.2. Frequency Domain

Figure 5 shows the Lomb–Scargle spectral analysis for different percentages of missing values and for each interpolation method on both time and duration. This figure shows that different interpolation methods introduce different deformations in the resulting power spectra. It is interesting to notice that performing no interpolation results in a flatter spectrum, more similar to a white noise.

In the frequency domain, the interpolation method that produces the least error is the quadratic on time (see Table 3). This figure shows that, as the amount of missing values increases, the no-interpolation approach tends to flatten the HRV spectrum, making it similar to the spectrum of white noise. Figure 5 also shows that cubic spline interpolation on time tends to dampen low frequencies while enhancing high frequencies; that cubic spline interpolation on duration tends to dampen all frequencies; and that quadratic interpolation on duration tends to enhance all frequencies. Finally, Figure 5 also shows that linear and quadratic interpolations on time and that nearest neighbour and linear interpolation on duration have minimal impact on both low and high frequencies, with quadratic interpolation on time having the least effect on all frequencies.

#### 3.2.3. Non-Linear Domain

SD1 does not require any interpolation to handle missing values, while SD2 needs quadratic interpolation on time to obtain reliable result (see Table 3). To give an explanation of these results, in Figure 6 we provide an example of the relationship between $RR-interval_{n}$ and $RR-interval_{n+1}$ (i.e., Poincaré plot) where SD1 and SD2 are extracted. This figure shows Poincaré plots obtained after interpolating missing RR-intervals by using different interpolation method on both time and duration. This figure shows that when the missing values were interpolated on time, the variability of SD1 reduced as the percentage of missing values increased, while the SD2 remain constant. Differently, the interpolation on duration introduce error on both SD1 and SD2 increasing their variability as the missing values increase. Finally, in this figure it can be seen that no-interpolation and quadratic interpolation on time introduced less error compared to the other method on SD1 and SD2, respectively.

## 4. Conclusions

In this work we quantify the expected error propagation of missing values in RR-intervals timeseries to HRV features, as a function of preprocessing interpolation approach, and amount of missing data. The main findings of this study is that the interpolation of missing values in RR-intervals timeseries on time (i.e., the heartbeats timestamps) produces more reliable HRV features estimations compared to interpolation on duration.

By using this preprocessing approach, the quantification of the expected error on HRV features caused by a huge amount of missing values (e.g., motion artifacts on a wrist-worn wearable device) can support better estimations of users’ well-being, by assessing their HRV features. This enables continuous passive monitoring of users’ cardiovascular activity in a non-obtrusive way, collecting data during their daily activities that could enable further research on preventative health.

A limitation of this study is the fact that we limited our focus on healthy subjects with normal heart activity, limiting the analysis to large amounts of missing values induced by motion artifacts, ignoring physiological phenomena such as ectopic beats.

Future studies will be useful for researcher and companies, which give insight into heart rate variability recorded by wrist worn IoT wearable devices, in order to better understand the potentiality of the data extracted from these devices to make inference about people heath status. Future work is needed to assess the influence of missing values simulated in accordance with motion and mechanical artifacts induced by external stimuli during data acquisition by using wrist worn IoT wearable devices. Finally, future works will also include the investigating the influence of missing values on HRV features on short timeseries (e.g., 2 min, 1 min and 30 s) and the identification of the shortest time required to obtain accurate estimation of users’ HRV features.

## Figures and Tables

**Figure 1 sensors-19-03163-f001:**
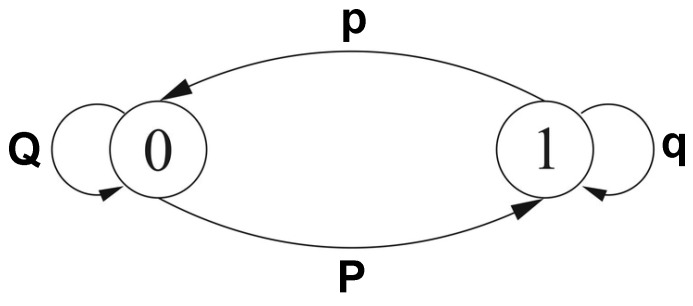
Gilbert model simulates burst-error with a two-state Markov chain (i.e., 0 and 1).

**Figure 2 sensors-19-03163-f002:**
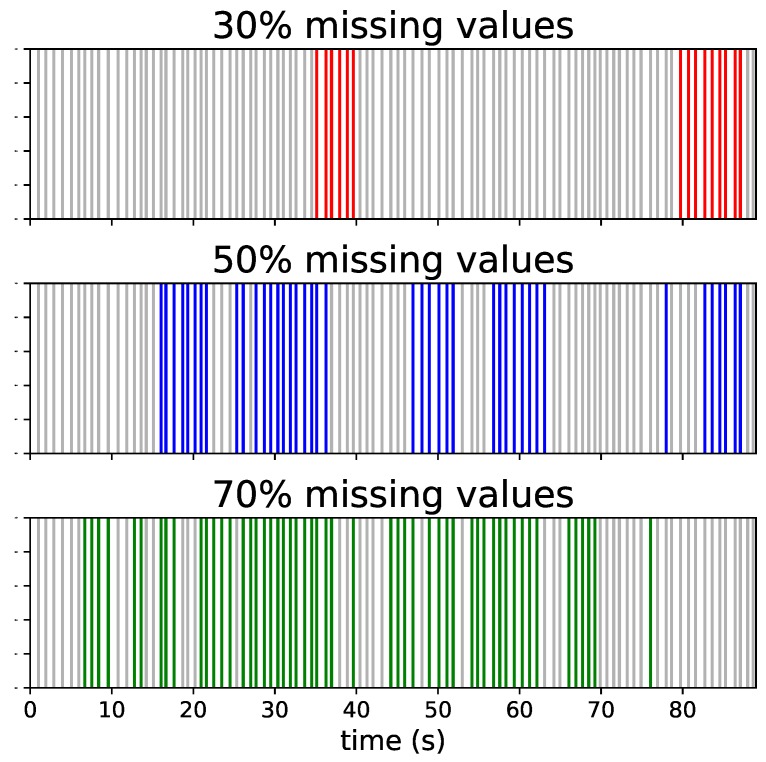
Examples of 30%, 50% and 70% of missing values created by Gilbert model. The colored lines refer to missing beats.

**Figure 3 sensors-19-03163-f003:**
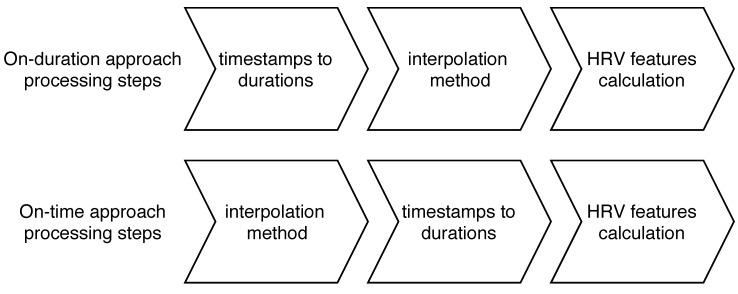
Processing steps in the on-time and in the on-duration approaches.

**Figure 4 sensors-19-03163-f004:**
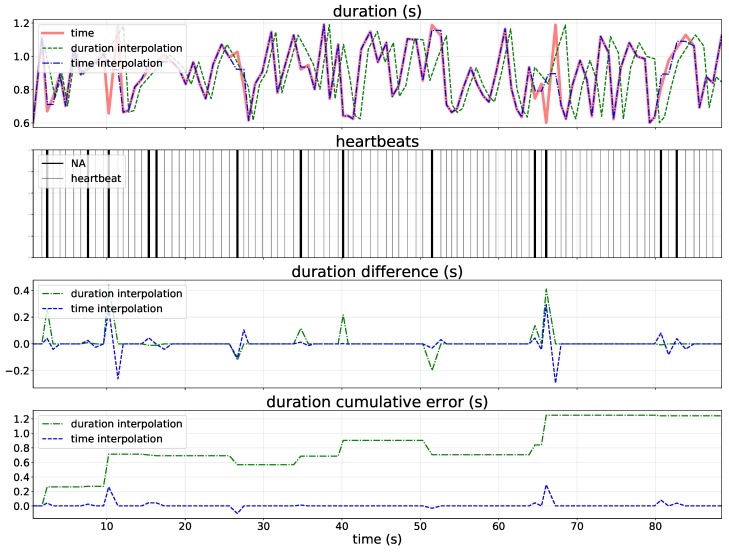
Difference between linear interpolation on time and duration. Red solid line refers to real variation of the timing between beats (RR-intervals) time series, green dashed line refers to the on-duration approach, and green dot dash line refers to the on-time approach.

**Figure 5 sensors-19-03163-f005:**
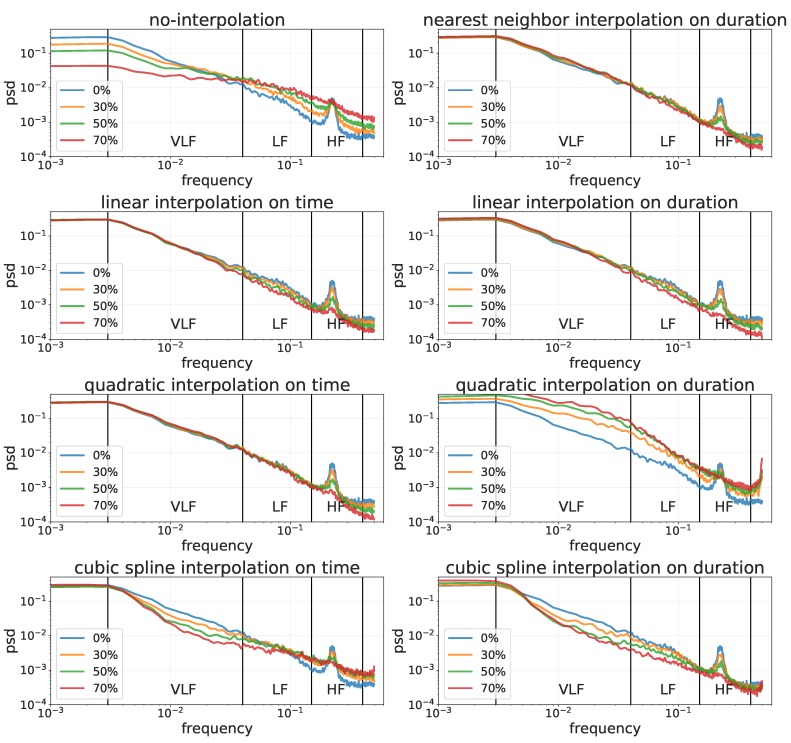
Frequency analysis of a user’s RR-intervals timeseries recorded in 5 min with different percentages of missing values (i.e., 0%, 30%, 50% and 70%) handled with different interpolation methods (i.e., nearest neighbor, linear, quadratic and cubic spline) on both time and duration.

**Figure 6 sensors-19-03163-f006:**
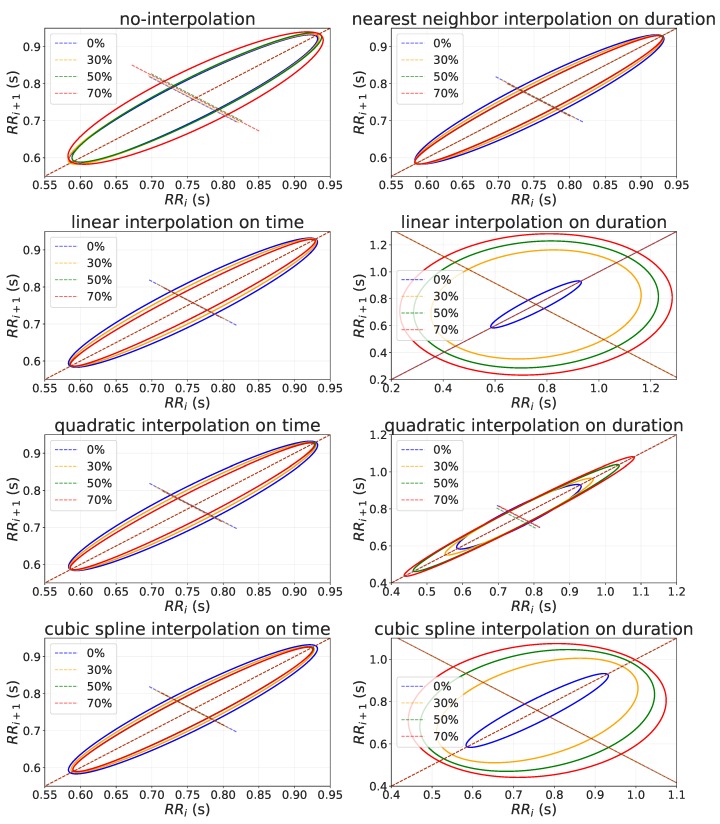
Poincaré plot of a user’s RR-intervals timeseries recorded in 5 min with different percentage of missing values (i.e., 0%, 30%, 50% and 70%) handled with different interpolation methods (i.e., nearest neighbor, linear, quadratic and cubic spline) on both time and duration.

**Table 1 sensors-19-03163-t001:** Difference between duration and time interpolation by using different approach (i.e., no-missing values, nearest neighbor, linear, quadratic, and cubic spline).

	Window Time (s)		RMSE (s)		RE (%)	
Interpolation	Time	Duration	Time	Duration	Time	Duration
No-missing values	90.11		—		—	
Nearest	—	91.95	—	0.096	—	5.11
Linear	90.11	91.83	0.075	0.090	3.70	4.86
Quadratic	90.11	92.13	0.084	0.107	4.35	5.83
Cubic spline	90.11	92.24	0.085	0.109	3.46	6.63

**Table 2 sensors-19-03163-t002:** Descriptive statistic of hart rate variability (HRV) features. Mean and 95% coefficient intervals (CI) are provided for all the feature.

HRV Features	Mean	95% CI
IBI (s)	0.78	[0.54, 1.11]
PNN50 (n)	8	[4, 16]
RMSSD (s)	0.039	[0.017, 0.36]
SD1 (s)	0.027	[0.012, 0.26]
SD2 (s)	0.077	[0.040, 0.25]
SDNN (s)	0.059	[0.017, 0.25]
VLF (s${}^{2}$)	0.87	[0.22, 4.15]
LF (s${}^{2}$)	0.477	[0.12, 5.57]
HF (s${}^{2}$)	0.28	[0.050, 3.024]
total power (s${}^{2}$)	1.91	[0.53, 21.44]
LF/HF (s${}^{2}$)	2.9	[1.2, 10.2]

**Table 3 sensors-19-03163-t003:** Best performing interpolation approach (i.e., with low RE) for each HRV feature in each percentage of missing values evaluated. The error in estimating HRV features is reported using RE and root mean squared error (RMSE).

		Interpolation			
Missing Values (%)	HRV	How	Method	RE (%)	RMSE
30	RMSSD (s)	No-interpolation		14.65	0.38
	SDNN (s)	Time	quadratic	9.42	0.34
	PNN50 (n)	No-interpolation		24.37	1.51
	SD1 (s)	No-interpolation		14.68	0.27
	SD2 (s)	Time	quadratic	8.57	0.47
	VLF (s${}^{2}$)	Time	quadratic	14.50	0.82
	LF (s${}^{2}$)	Time	quadratic	26.87	2.01
	HF (s${}^{2}$)	Time	quadratic	32.18	4.48
	LF/HF (s${}^{2}$)	Time	cubic	41.39	1.73
	total power (s${}^{2}$)	Time	quadratic	17.16	6.26
50	RMSSD (ms)	No-interpolation		23.13	0.76
	SDNN (s)	Time	quadratic	15.47	0.41
	PNN50 (n)	No-interpolation		39.01	2.35
	SD1 (s)	No-interpolation		23.18	0.54
	SD2 (s)	Time	quadratic	13.49	0.49
	VLF (s${}^{2}$)	Time	quadratic	23.72	0.40
	LF (s${}^{2}$)	Time	quadratic	42.42	1.12
	HF (s${}^{2}$)	Time	quadratic	52.56	2.48
	LF/HF (s${}^{2}$)	Time	cubic	58.07	2.26
	total power (s${}^{2}$)	Time	quadratic	27.59	3.96
70	RMSSD (s)	No-interpolation		34.37	0.91
	SDNN (s)	Time	quadratic	22.76	0.47
	PNN50 (n)	Time	linear	63.90	3.88
	SD1 (s)	No-interpolation		34.46	0.59
	SD2 (s)	Time	quadratic	19.19	0.51
	VLF (s${}^{2}$)	Time	quadratic	29.73	0.52
	LF (s${}^{2}$)	Time	quadratic	56.41	1.45
	HF (s${}^{2}$)	Time	quadratic	72.98	3.34
	LF/HF (s${}^{2}$)	Time	cubic	72.07	2.80
	total power (s${}^{2}$)	Time	quadratic	72.07	5.27

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
