# Peer review of "Analysis of the Impact of Interpolation Methods of Missing RR-intervals Caused by Motion Artifacts on HRV Features Estimations"

_sensors, 2019, doi:10.3390/s19143163_

Round 1

Reviewer 1 Report

The authors presented a method for finding heart rate at the presence of the possible missing beats. Although there are incorrect and exaggerations throughout the manuscript (e.g. Abstract L.1 , L.2 "s that are commonly 2 worn by a large part of the population during their daily life", the method sounds a contribution, to some extent, but there are serious criticisms to the presented study  in terms of the applicative contents. One of the important shortcoming of the investigation is the presence of atrial fibrillation and premature heart beat on a ECG recording that should be detected by using the wearable sensors. It is important the authors include such conditions in their dataset and investigate the performance of the presented methods.

Author Response

Response to REVIEWER 1 Comments

The authors presented a method for finding heart rate at the presence of the possible missing beats. Although there are incorrect and exaggerations throughout the manuscript (e.g. Abstract L.1 , L.2 "s that are commonly 2 worn by a large part of the population during their daily life"). The method sounds a contribution, to some extent, but there are serious criticisms to the presented study  in terms of the applicative contents. One of the important shortcoming of the investigation is the presence of atrial fibrillation and premature heart beat on a ECG recording that should be detected by using the wearable sensors. It is important the authors include such conditions in their dataset and investigate the performance of the presented methods.

Author response: We thank reviewer 1 for the useful comments. First of all, we carefully modified sentences containing all the erroneous information and exaggerations. Moreover, we added some details about missing values throughout the paper highlighting that the wrist-worn wearable devices produce several inconsistent RR-intervals induced not only by ectopic beats (e.g., atrial fibrillation and premature heart beat), but the majority of them are due to motion and mechanical artifacts induced by external stimuli. However, the reason why the missing values are introduced into the RR-intervals time series is off topic on our paper. As a matter of fact, we are interested in investigating the effect of two different interpolation strategies (i.e., on Time and on Duration) on time series with high percentage of missing values (from 30 to 70% of missing values).

Reviewer 2 Report

This paper evaluates and compares various interpolation strategies in order to address the problem of missing RR-intervals in HRV analysis. By using RR interval data obtained from the NSR2DB, the authors conclude that quadratic interpolation on time is the best approaches to reconstruct the missing RR-intervals.

Although the paper is clearly organized and technically focused, some drawbacks do not allow to appreciate the soundness of the proposed approaches. The main problems of this paper are listed below.

The section "1.2. Related work" is thin and out of date. The authors should include more recent literature, published in 2018 and 2019, clearly focusing on interpolation strategies in time and duration which is the topic of this paper. The scientific relevance of interpolation strategies in both time and duration should clearly emerge from cited recent works.

The paper contains lots of assumed knowledge and descriptions of some sections lack in detail or are not clear. In section "2.2. Missing values interpolation", duration interpolation and time interpolation methodologies should be detailed explained. Additional figures, schema, pseudocode and/or simpler description by relating to similar works (published elsewhere) should be provided.

All metrics reported in Table 1 should be numerically defined (e.g., IBI, LF, HF, etc.). To avoid confusing the reader, abbreviations should be referred in the same way, using always capital/small letters.

Some figures can be improved. Figure 3 would be more useful in section "2. Materials and Methods" to show differences between on-time and on-duration interpolation methods. At this purpose, in the first (topmost) plot, miss RR-interval values should be included, highlighting how they are reconstructed by using the two interpolation methods.

In Figures 3 and 4, x and y axis labels and units should be placed along each axis.

In the 4x2 plot grid of Figure 5, plots related to nearest-neighbor interpolation on duration, and linear, quadratic and cubic-spline interpolation on time are virtually the same. Thus, it is not clear how one can conclude that no-interpolation and quadratic interpolation on time introduce less error. This can be addressed by either zooming or using a different type of plot.

There are problems with broken cross references in the paper. Some of these have been highlighted below:

- P. 7, L. 224: "in Table ??, ??"

- P. 9, L. 254: "Appendix ??"

Author Response

Response to REVIEWER 2 comments

This paper evaluates and compares various interpolation strategies in order to address the problem of missing RR-intervals in HRV analysis. By using RR interval data obtained from the NSR2DB, the authors conclude that quadratic interpolation on time is the best approaches to reconstruct the missing RR-intervals. Although the paper is clearly organized and technically focused, some drawbacks do not allow to appreciate the soundness of the proposed approaches. The main problems of this paper are listed below.

Author response: We thank reviewer 2 for the useful comments. We modified the text in accordance with the reviewer’s comments listed above. All the answers were placed after each question.

Question 2.1. The section "1.2. Related work" is thin and out of date. The authors should include more recent literature, published in 2018 and 2019, clearly focusing on interpolation strategies in time and duration which is the topic of this paper. The scientific relevance of interpolation strategies in both time and duration should clearly emerge from cited recent works.

Author response: The reason why we do not provide literature about interpolation on Time is due to the fact that to the best of our knowledge this is the first study that suggest to interpolate missing values on Time (i.e., the timestamps when the heartbeats happen) instead of interpolating on Duration (i.e., the duration of the heartbeats). In the “Related work” section we provide only the most important previous papers that have proposed to interpolate the missing RR-intervals on Duration (i.e., the strategies used to handle missing values by now). The most recent papers are not focused on assessing the effect of interpolation approach on missing values, but these researchers used it to handle ectopic beats in order to obtain time series where it is possible to accurately estimate HRV parameters able to answer the research question of these papers. We do not report the results of these papers because they are not in line with our topic.

We added a sentence in the “Related work” section to explicitly mention that we were not able to find any previous work studying interpolation on time.

Question 2.2. The paper contains lots of assumed knowledge and descriptions of some sections lack in detail or are not clear. In section "2.2. Missing values interpolation", duration interpolation and time interpolation methodologies should be detailed explained. Additional figures, schema, pseudocode and/or simpler description by relating to similar works (published elsewhere) should be provided.

Author response: For both the on-Time and on-Duration approaches we provide a detailed description of the processing steps and added a diagram that shows the differences in the order of those steps for the two approaches.

Question 2.3. All metrics reported in Table 1 should be numerically defined (e.g., IBI, LF, HF, etc.). To avoid confusing the reader, abbreviations should be referred in the same way, using always capital/small letters.

Author response: We modified all the abbreviation throughout the paper using always capital letters. We add a new table (i.e., new Table 1 in the revised paper. The Table 1 became table 2 in the revised version of the paper) with mean and 95% coefficient intervals for all the features provided on the text. In Table 2 of the revised paper version, we add the unit of measurement for all the features.

Question 2.4. Some figures can be improved. Figure 3 would be more useful in section "2. Materials and Methods" to show differences between on-time and on-duration interpolation methods. At this purpose, in the first (topmost) plot, miss RR-interval values should be included, highlighting how they are reconstructed by using the two interpolation methods.

Author response: We move the description, Figure and Table describing the differences between on-time and on-duration interpolation methods in Materials and Methods section. In the first plot we provide three time series: (i) the RR-intervals time series without missing values; (ii) the RR-intervals time series with missing values interpolated on Time; (iii) the RR-intervals time series with missing values interpolated on Duration. Hence, the missing RR-intervals are already provided in this plot on the time series (i). Moreover, the algorithm used to interpolate missing RR-intervals on Time and Duration are provided in the Algorithm 1 and 2. The other graph presented in Figure 3 provided more detailed information about error introduced interpolating missing values on Time and Duration.

Question 2.5. In Figures 3 and 4, x and y axis labels and units should be placed along each axis.

Author response: We have modified Figure 3 and 4 in accordance with this comment.

Question 2.6. In the 4x2 plot grid of Figure 5, plots related to nearest-neighbor interpolation on duration, and linear, quadratic and cubic-spline interpolation on time are virtually the same. Thus, it is not clear how one can conclude that no-interpolation and quadratic interpolation on time introduce less error. This can be addressed by either zooming or using a different type of plot.

Author response: We thank the reviewer for his suggestion. We rescaled each plot to zoom the results highlighting the error difference for each method, strategies and percentage of missing values.

Question 2.6. There are problems with broken cross references in the paper. Some of these have been highlighted below:

- P. 7, L. 224: "in Table ??, ??"

- P. 9, L. 254: "Appendix ??"

Author response: We are sorry for the mistake. These are misprints from previous draft of the paper. We deleted these Tables and Appendices just before the submission process and we forgot to delete these sentences. We now deleted it.

Round 2

Reviewer 1 Report

The authors made a great effort to improve the manuscript in terms of the writing skill. However, lack of the abnormal ECG conditions restricts applicability of the presented method and provides an unfair conclusion about the comparisons. The improvement of the method in comparison with the baseline may not be conclusive in the pathological conditions which can be an important focus for such the technologies.

Author Response

Thank you for your appreciation on our effort to improve the manuscript. 

To address your concerns we better specified the scope of our study by changing the title, the abstract and the conclusions. We clearly indicated that we limit our focus to preprocessing of RR data with large amounts of missing data caused by motion artifacts. 

We took in consideration your intuition during the development of our future  research design. We already scheduled future works to investigate the sensitivity of our interpolation methodology to discriminate between normal and abnormal ECG conditions. However, for the scope of this study, we wanted to understand how much error is introduced by missing RR-intervals on HRV feature estimation using different interpolation approaches. To this aim, the introduction of abnormal ECG could introduce more error due to physiological condition that could have an effect on HRV parameters. Hence, we chose to analyse the validity of different interpolation approaches only on ECG derived from non pathological conditions, limiting our experiment to data from healthy subjects.

Reviewer 2 Report

This paper evaluates and compares various interpolation strategies in order to address the problem of missing RR-intervals in HRV analysis. By using RR interval data obtained from the NSR2DB, the authors conclude that quadratic interpolation on time is the best approaches to reconstruct the missing RR-intervals.

The author has made all necessary amendments to previous reviewer comments so the manuscript is now ready for acceptance.

Author Response

We would like to thank the reviewer for his helpful suggestions in his previous revision that allowed us to reach the level for the publication in Sensors.

Round 3

Reviewer 1 Report

I have no further comments.